# Using All-Atom Potentials to Refine RNA Structure Predictions of SARS-CoV-2 Stem Loops

**DOI:** 10.3390/ijms21176188

**Published:** 2020-08-27

**Authors:** Christina Bergonzo, Andrea L. Szakal

**Affiliations:** 1Institute for Bioscience and Biotechnology Research, National Institute of Standards and Technology and University of Maryland 9600 Gudelsky Drive, Rockville, MD 20850, USA; 2National Institute of Standards and Technology 100 Bureau Drive, Gaithersburg, MD 20899, USA; andrea.szakal@nist.gov

**Keywords:** structure refinement, molecular dynamics, RNA stem loops, discontinuous transcription

## Abstract

A considerable amount of rapid-paced research is underway to combat the SARS-CoV-2 pandemic. In this work, we assess the 3D structure of the 5′ untranslated region of its RNA, in the hopes that stable secondary structures can be targeted, interrupted, or otherwise measured. To this end, we have combined molecular dynamics simulations with previous Nuclear Magnetic Resonance measurements for stem loop 2 of SARS-CoV-1 to refine 3D structure predictions of that stem loop. We find that relatively short sampling times allow for loop rearrangement from predicted structures determined in absence of water or ions, to structures better aligned with experimental data. We then use molecular dynamics to predict the refined structure of the transcription regulatory leader sequence (TRS-L) region which includes stem loop 3, and show that arrangement of the loop around exchangeable monovalent potassium can interpret the conformational equilibrium determined by in-cell dimethyl sulfate (DMS) data.

## 1. Introduction 

Severe acute respiratory syndrome coronavirus 2 (SARS-CoV-2), a betacoronavirus which emerged in December 2019, is an ongoing pandemic [1]. Characterization of the viral ribonucleic acid (RNA) has been ongoing and is inspiring a multi-faceted approach to combating the virus. Higher order structures in the viral RNA present attractive targets for small molecule anti-SARS-CoV-2 drugs [2]. Additionally, to aid in quantification of SARS-CoV-2 viral RNA, the National Institute of Standards and Technology (NIST) has released research grade test material containing synthetic RNA for use in development and validation of reverse transcription polymerase chain reaction (RT-PCR) based diagnostic assays for SARS-CoV-2 [3,4]. To this end, fragment selection for quantification or for targeting could be informed by prediction of RNA higher order structure, which could impact the reverse transcription step of the assay. 

Specifically, many questions arise around higher order structure in the 5′ untranslated region (5′UTR) of the viral genome, since its structural conservation has been noted throughout genera in the coronavirus family, and the cis-acting regulatory elements proven to be functionally important [5]. Stem loops 2 and 3 (SL2 and SL3, respectively) higher order structures are of interest, since SL2 structure is conserved throughout coronaviruses and SL3 contains the conserved transcription regulatory leader sequence (TRS-L), involved in discontinuous transcription [6]. 

Multiple studies of the single-stranded coronavirus genome show the RNA is a template for synthesis of the full length genome, as well as subgenomic negative sense RNAs which are produced by discontinuous transcription and regulate protein expression [7,8]. Long-range RNA–RNA interactions govern these mechanisms [9]. Discontinuous transcription is mediated by base pairing between the 5′UTR TRS-L sequence and the complement of the transcription regulatory body sequence TRS-B at the 5′ ends of open reading frames (ORFs) [10,11]. In betacoronaviruses the TRS-L sequence is 5′-UCUAAAC-3′. The SL3 TRS-L region has been proposed to be non-structured in Middle East respiratory syndrome related coronavirus (MERS-CoV) and murine coronavirus (MHV), but adopt a stem loop structure in SARS-CoV and bovine coronavirus (BCoV) [6], depending on the thermodynamic stability of the nucleotides flanking the TRS-L in forming a helix. 

In work published by the Das group in Rangan et al. [12,13], the 2D and 3D structures of the 5′UTR, 3′UTR and frame shifting element (FSE) domains from SARS-CoV-2 viral RNA are predicted. Higher order structures are reported in two forms: as individual fragments of each domain’s RNA which have some predicted secondary structure, and as complete elements showing the global orientation of each secondary structure element. Fragment Assembly of RNA with Full-Atom Refinement (FARFAR2), a Rosetta program used to guide 3D RNA structure prediction based on homology modeling to known/solved motifs, was used to make these predictions about the higher order structure in SARS-CoV-2 RNA [14]. Structures created at low resolution are subsequently refined using an all-atom physics based force field [15]. This work also incorporated chemical mapping experiments to inform the 3D structure predictions. 

De novo structure prediction for RNA has yet to achieve the wide popularity that protein structure prediction has. Many limitations exist, including limited high resolution 3D structural data for RNAs in general, which limits the pool of structure fragments from which to pull information [16]. Additionally, though conformations are predicted, static structures provide little to no insight into the dynamics of the system. Molecular dynamics simulations employing all-atom force fields can be used to predict the time-dependent motions involved in these RNAs [17], as well as their thermodynamic probabilities for particular conformations, factors which are critical to understanding higher order structure ensembles intended to be used as targets for small molecule drugs. Additionally, the inclusion of explicit water and ions and the implementation of particle mesh Ewald for modeling long-range electrostatic interactions returns results more consistent with experimental data than a continuum representation, or more primitive treatment of electrostatic terms [16,18,19]. 

In this work, we present refined 3D structure ensembles for the highly conserved SL2 consistent with previously determined solution state experimental NMR data from SARS-CoV-1 [20]. We also refine the proposed SL3 SARS-CoV-2 hairpin loop structures and show that transient K^+^ metal ion association can partially stabilize the TRS-L sequence, suggesting a mechanism for transitioning between the non-structured and higher order structured regions necessary for transcription regulation. 

## 2. Results

### 2.1. MD Refinement of SL2 

As the authors acknowledge in the original prediction, the FARFAR2 structures of SL2 reflect a limitation in the Rosetta energy function which charges two bases of the loop (C and G) a desolvation penalty without rewarding stacking, and structures are also affected by limitations in sampling [13]. In theory, both of these limitations can be addressed by using all-atom molecular dynamics potentials, which include explicit solvent, and extensive sampling with replicates to gauge statistics. To understand the deviations in structures that arise from different treatments of an RNA (FARFAR2 homology modeling and structure prediction using fragment assembly, vacuo modeling using database potential, and molecular dynamics using all-atom potentials including explicit solvent), we took the representative cluster centers from each of the ten reported SARS-CoV-2 SL2 FARFAR2 clusters, and the top ten lowest energy structures from the SARS-CoV-1 NMR ensemble [20], standardized the sequence by removing the 3′A overhang, and solvated them to run molecular dynamics using modern force fields. FARFAR2 starting structures had a mean pairwise root mean square deviation (RMSD) of 1.18 Å, and NMR starting structures had a mean pairwise RMSD of 0.3 Å.

#### 2.1.1. Refinement of SL2

Figure 1 shows the 3D structures and sequences of the FARFAR2 (left) and NMR (right) predicted SL2 RNA. Simulations of ten starting structures for each system were each run for 1–3 microseconds. We were looking for validation of the NMR model, as well as rearrangement of the FARFAR2 prediction to sample structures closer to the NMR model. Appendix A shows time-dependent RMSDs for each of these twenty simulations, and indicate long, stable trajectories in simulations starting from the NMR models with an average RMSD to the starting structures of about 2 Å for loop residues and 1 Å for stem residues. Simulations starting from FARFAR2 predicted models have 3–7 Å deviations from their starting loop structures. Over three microsecond simulations for the FARFAR2 structures, none of the loops remain in their initial predicted conformations. Nearly all move on to sample alternate structures, though a few sample their original starting structure.

Appendix A report the formation of the canonical CUUG C-G loop base pair by measuring the central Watson–Crick hydrogen bond between C N3 and G H1, for the NMR (C47-G50) and FARFAR2 (C50-G53) set of simulations. For the NMR set, a significant amount of structures maintains the initial CG configuration, with slight deviations to longer hydrogen-bonding distances, indicating some fluctuations in the loop. For the FARFAR2 set, a significant amount of structures keeps the originally predicted CU base pair, measured by C50 N3 to U54 H3 distance and reported in Appendix A. However, there is much more sampling of long C50 N3 to U54 H3 distances, and additionally some sampling of Watson–Crick hydrogen bond distances between C50 and G53. We predict that rearrangement to the CG base pair of the CUUG tetraloop shifts the pentaloop structures closer to those measured in the solution state NMR.

#### 2.1.2. Validation of SL2 Prediction

To validate our prediction of the new CUUG-like FARFAR2 SL2 conformations from molecular dynamics (MD), we conducted analysis on several ensembles: the starting models from the deposited 2L6I NMR, the MD simulations starting from those deposited models (NMR MD), the MD simulations starting from FARFAR2 models which sample the central CG hydrogen bond (FAR CG) and the MD simulations starting from FARFAR2 models which sample the central CU hydrogen bond (FAR CU). The criteria for filtering the FARFAR2 simulation into the FAR CG ensemble was a C50@N3-G53@H1 distance less than 3.0 Å, and the criteria for filtering the FARFAR2 simulation into the FAR CU ensemble was a C50@N3-U54@H3 distance less than 3.0 Å. The central CG hydrogen bond is sampled in all simulations in varying amounts (Appendix A). We evaluated each of these ensembles against the deposited NMR data, including nuclear Overhauser effect measurements (NOEs) and chemical shifts. 

Figure 2 shows the results of NOE distance analysis for the structure ensembles. Independent NOE distances were calculated for each structure and r^6^ averaged over the structure ensembles. Those averages are reported here for any NOE violation greater than 0.2 Å. Global NMR numbering is used for all ensembles in this Figure. Figure 2, top, shows the calculated proton distances, and reports on the strength of the NOE which is violated. Figure 2, bottom, shows the difference between the calculated and measured NOE distances, and reports on the magnitude of the deviation. The closest values to the measured NOE distances are from the starting ensemble, while the NMR MD ensemble shows only three violations >1.0 Å, which all occur in the loop region and involve the solvent exposed (and highly dynamic) U51. The FAR CG ensemble shows many more violations, in the loop region especially, but indicates overall good qualitative agreement with the NMR data. Deviations involve NOEs from residues U49 and U51. While U51 is expected to be dynamic, U49 should stack on the C of the C-G Watson–Crick base pair in the CUUG loop. While this stacking does occur, there is an accompanying shift in the sugar pucker and chi distribution that contributes to violation of several NOEs (Appendix A). The FAR CU ensemble shows violations of almost every loop NOE distance, sometimes up to 5.0 Å from the measured distances. Table 1 reports the chi squared goodness of fit statistic for each ensemble’s NOEs, and quantify the results reported by NOE analysis, namely that the fit to experimental data for NMR MD ensemble ≅ FAR CG ensemble << FAR CU ensemble.

The six backbone dihedrals (α, β, γ, δ, ε, and ζ) for the SL2 CUUGU loop residues show good agreement between the starting ensemble, NMR MD ensemble, and FAR CG ensemble, while the FAR CU ensemble shows the largest deviations from experiment and within the ensemble itself (Appendix A). Pucker and chi distributions, presented as histograms in Appendix A, show the split between North (C3′endo) and South (C2′endo) sugar puckering. The reported puckering for the loop from the NMR MD ensemble agrees with the reported NMR data. For most residues, the FAR CG ensemble has some population that overlaps the NMR MD ensemble but includes a significant population in the alternate pucker conformation. The FAR CU ensemble has a higher population of puckers that do not match the NMR data in some cases: U48, G50, and U51 (in NMR numbering). There is still significant overlap between the FAR CU and FAR CG ensembles, which reflects the difficulty of rearranging puckering on these timescales. Chi distributions remain consistently in the Anti population across the loop residues, except for G50 FAR ensembles, both of which sample a significant amount of Syn population.

Appendix A shows the comparison to proton chemical shift data, predicted from each structure in a pared down ensemble (1000 structures per NMR MD, FAR CG, and FAR CU ensembles taken at even intervals across all trajectories), with averages and standard deviations calculated over all 1000 structures. The predictions are split by shift type based on their expected ppm values: base H2/H6/H8, base H5, sugar H1′, and sugar-backbone H2′, H3′, H4′, H5′, and H5″. The overall trend matches the NOE distance evaluation: the starting ensemble and NMR MD sets have the highest correlation to the measured values (reported as Pearson correlation coefficient of a linear fit), and the lowest standard error of the regression coefficient (reported as SEC), indicating the best agreement to the measured data. Interestingly, the flexibility sampled by the MD ensemble results in better fit to the data than the starting structures, at least in terms of fulfilling base chemical shifts. The FAR CG ensemble is the next-closest fitting ensemble, while the FAR CU ensemble shows the widest standard deviation in predicted shifts, and overall worst agreement to the measured NMR chemical shift data for this sequence/loop.

### 2.2. MD Refinement of SL3 

Since MD was used to generate SL2 structures showing better agreement with experimental NMR data, we hypothesize it can be used to refine FARFAR2 predictions of SL3 in a similar manner. Ten simulations were run, starting from the top structures from ten clusters of predicted SL3 stem loops (Appendix A). Most simulations deviated in the loop region from the initial structures while the stem remained largely intact in an A-form helix. However, for at least three simulations unfolding of the helix was observed, as well as transient base pair opening in many others. Cluster analysis was performed for the loop region plus two base pairs of the stem to act as an anchor point over all simulated structures, and a representative structure accounting for 30% of all sampled loop structures was found. Figure 3 compares the top de novo predicted secondary structure of SL3, the TRS-L region in the 5′UTR of the SARS-CoV-2 viral RNA, and the representative structure from cluster analysis of MD simulations performed starting from the FARFAR2 predictions. Cluster metrics are reported in Appendix A, and Appendix A shows the representative structures with ten structures from each cluster trajectory overlapped for the top four most populated clusters. 

#### 2.2.1. Predicted SL3 Structures

The representative structure from MD simulations shows a rearrangement of the seven-membered loop region compared to the FARFAR2 predictions. Starting FARFAR2 structures predict hydrogen bonds between U65 and C71 via their O2 and H41 atoms with, additional stabilization through hydrogen bonding between A69 O2′ and U65 H3/ C71 H42. The C66 nucleobase is rotated 90° out of the Watson–Crick base pairing plane, and A69/A70 are co-located in the same plane. The four base pair stem is A-form. In the structure reported from MD simulations, the transient U5-C11 base pair interactions are intact, with the absence of the additional stabilizing contribution from A9 ribose hydroxyl group. On top of this “base pair” an AC “base pair” forms through a hydrogen bond between C6 O2 and A10 H62 atoms. Finally, the loop is capped by a solvent exposed U and the A8-A10 stretch stacks in a helical conformation. A key interaction in the predicted SL3 motif is the organization of the loop around a K^+^ ion, shown as an orange sphere in Figure 3. 

#### 2.2.2. The Role of the Solvent and Ions in SL3

At the center top of the SL3 seven-membered loop predicted by MD simulation, a K^+^ ion coordinates residues C6, A8, A9, and A10. Figure 4 shows the overlap of 100 frames from the top cluster with K^+^ ion density at 80% occupancy. The bar graph in Figure 4 shows the Fraction Occupancy by residue for SL3. Specifically, coordination with K^+^ occurs through contacts with C6 O2 and N3, A8 and A9 phosphate oxygen OP2, and A9 N7 atoms in this binding site. Measurements of the closest K^+^ ions to the binding site ions show exchanges between bound ions and free ions in solution (Appendix A).

## 3. Discussion

In the 5′UTR of the SARS-CoV-2 genome, SL3 covers the TRS-L region. SL2 is located directly 5′ to SL3 and is a conserved stem loop throughout coronavirus genera. Understanding the higher order structures in SARS-CoV-2 RNA can promote drug discovery efforts targeting RNA secondary structure motifs, and the importance of generating reasonable predictions of 2D and 3D structures via the work of Rangan et al. cannot be overstated. In addition to rapid structure-based drug design efforts and measurement efforts surrounding live virus testing, each of these short regions poses different and interesting questions in current practices for modeling higher order RNA structure.

The predicted structure of SL2 by the FARFAR2 program did not organically reproduce the experimentally determined structure determined by solution state NMR for SL2 of SARS-CoV-1 [20]. Significant deviation from experimental measurements are reported for structures which maintain the FARFAR2 predicted C–U interaction, while those which rearrange to form the C–G interaction seen in CUYG-like tetraloops (where Y denotes a pyrimidine base) reproduce this data to a much higher degree (Figure 2, Table 1, Appendix A). Additionally, structures which rearrange to this hydrogen bonding pattern show consistency with predicted backbone dihedral values averaged from simulations of the NMR models. Although the majority of structures with the C–G Watson–Crick hydrogen bond reproduce the experimental data to an acceptable degree, it remains both an imperfect metric to gauge rearrangement of the loop to the CUYG tetraloop. Structures began to deviate from their initial conformations after about 500 ns of dynamics (Appendix A). Deviations in the chi and pucker distributions from known CUYG tetraloop ranges are still present, and speak to the longer timescales of simulation needed to rearrange and refold tetraloops [21,22].

There is some discrepancy in SL3 higher order structure prediction between coronaviruses. No structure is predicted for MERS-CoV and MHV, while a stem loop containing the TRS-L sequence, which maintains a dynamic ensemble consisting of folded and unfolded regions, is predicted for BCoV and SARS-CoV [6]. Part of the biological role of TRS-L is to pair with the TRS-B region, indicating a great likelihood that this region needs to be unstructured to fulfil this role, and further proposing this is accomplished by a conformational switch mechanism [11]. Molecular dynamics simulations performed in this work characterize the dynamics of this important functional loop. We propose a new model in which ion binding and unbinding contributes to intermediate stability of this piece of RNA. The specific interactions are reproducible and are shown to lightly tether the hairpin loop together. However, these interactions are K^+^ dependent allowing multiple transitions between “stable” loop structures. Since K^+^ is easily hydrated, increased dynamics in the loop may easily be promoted through binding/unbinding events [23,24]. K^+^ presents a transient stabilization point for this motif, the absence of which would allow unfolding of the already unstable helical stretch. Indeed, even in these short simulations the helix shows some evidence of unfolding at 300 K.

Agreement with in-cell chemical modification data further supports a dynamic 3D model [25]. In this work, in-cell dimethyl sulfate (DMS) chemical probing coupled with mutational profiling through next-generation sequencing (DMS-MaPseq) was performed. DMS methylates unpaired adenine and cytosine bases at their Watson–Crick faces, after which the RNA is fragmented, reverse transcribed, and sequenced to read the amounts of modifications made, as a reporter on nucleotide flexibility. Appendix A maps the fluctuations of each residue in SL2 and SL3 onto their measured DMS reactivities and shows high flexibility in SL3 and low flexibility in SL2. Taken together, these qualitative pieces of evidence present a cohesive model for SL3 structure and function.

## 4. Materials and Methods 

Starting structures for SL2 simulations were taken from Rangan et al. [13] and from deposited NMR models 2L6I in the protein data bank [20]. These SL2 structures were standardized to the SARS-CoV-2 sequence by removing the 3′A overhang from the deposited 2L6I models. For structures taken from de novo prediction, the top structure of each of the top ten clusters was used as a starting structure for a simulation. For structures taken from 2L6I, the top ten deposited models were each used as a starting structure for a simulation. Starting structures for SL3 simulations were taken from Rangan et al. [13]. Each structure was built using Link-Edit-and-Parm (LEaP) in Amber [26] using the LJbb force field described in Bergonzo and Cheatham [27]. Optimal point charge (OPC) water [28] was used to solvate the RNA in a truncated octahedron unit cell with at least 10 Å buffer from solute atoms. K^+^ atoms were added to neutralize charge, and an additional 1 K^+^ ion and 1 Cl^-^ ion were added to match 10 mmol/L concentrations used in the solution NMR experiment [29]. Ion positions were randomized 6 Å from the solute RNA and 4 Å from each other. 

Minimization and equilibration were run for each of the 30 structures as previously described [30]. Briefly, after initial minimization and heating in a canonical (NVT) ensemble, steepest descent and conjugate gradient minimization steps were followed by isobaric-isothermal (NPT) equilibration iteratively with decreasing positional restraints on the RNA. The last step of equilibration was run for 50 ps at constant pressure with no restraints. Production dynamics were run using hydrogen mass repartitioning, in which hydrogen masses are increased by redistributing mass from the heavy atoms to which they are covalently bound, resulting in the ability to use a 4 fs timestep [31] in conjunction with SHAKE constraints on hydrogen atoms [32]. Amber graphics processing unit (GPU)-accelerated particle mesh Ewald molecular dynamics (PMEMD) was used to run simulations [33,34] in an NVT ensemble. Temperature was set to 300 K to match experimental conditions and regulated using a Langevin thermostat, with a collision frequency of 2 ps^−1^ and using random seeds for initial velocity assignments to prevent synchronization artifacts [35,36]. Simulations of all stem loops (SL2 and SL3) were run for 1–3 microseconds. Simulations were analyzed using CPPTRAJ [37,38]. Cluster analysis was performed on the mass weighted residues 3–13 of SL3 using a kmeans algorithm with a random seed. Visualizations were prepared using Visual Molecular Dynamics (VMD) [39]. Proton chemical shifts were calculated using SHIFTS [40]. 

## 5. Conclusions

In refining SARS-CoV-2 SL2 RNA from FARFAR2 predicted 3D structures, we have shown that advanced all-atom potentials can improve 3D RNA de novo predictions. The simulation results reported in this work demonstrate the rearrangement of homology or fragment-based RNA secondary structure predictions to experimentally validated structures. This demonstrates adequate accuracy of the MD force field in differentiating experimentally determined structures from high quality de novo prediction—as the FARFAR2 structures rearrange during long unbiased MD simulations, we can say that the NMR ensemble is closer to the global minimum of the MD force field. Further, the results show that long unbiased MD simulations account for known inaccuracies in the FARFAR2 energy function. The refined SL3 structure prediction provides a reasonable hypothesis for linking the RNA secondary structure to the known TRS-L sequence function, accounting for the dynamics of this sequence which are missed by de novo models.

## Figures and Tables

**Figure 1 ijms-21-06188-f001:**
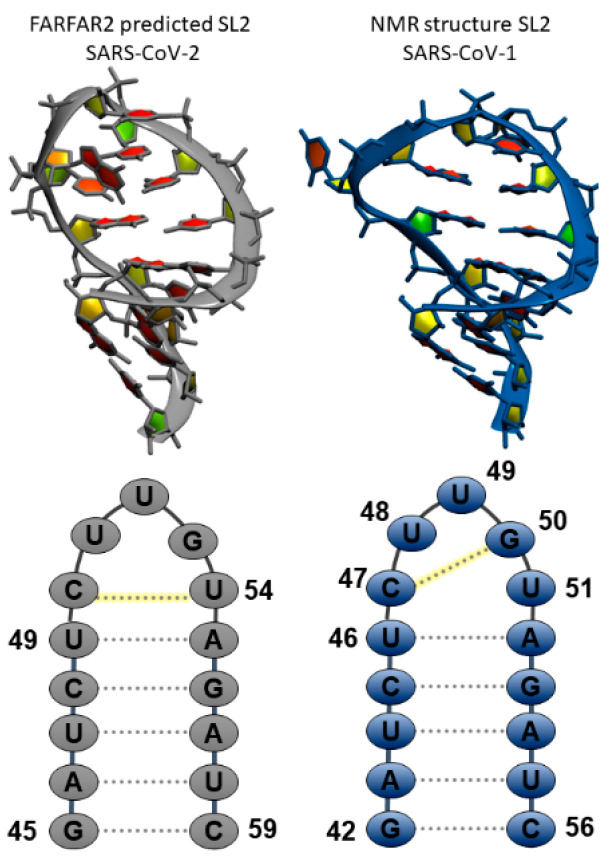
Comparison of predicted stem loop 2 (SL2) structures from (**left**) de novo tertiary structure prediction Fragment Assembly of RNA with Full-Atom Refinement (FARFAR2) and (**right**) NMR structure for SL2 in Severe acute respiratory syndrome coronavirus 1 (SARS-CoV-1). Sequences with global numbering and deviations in the secondary structure preferences are highlighted in the bottom frames.

**Figure 2 ijms-21-06188-f002:**
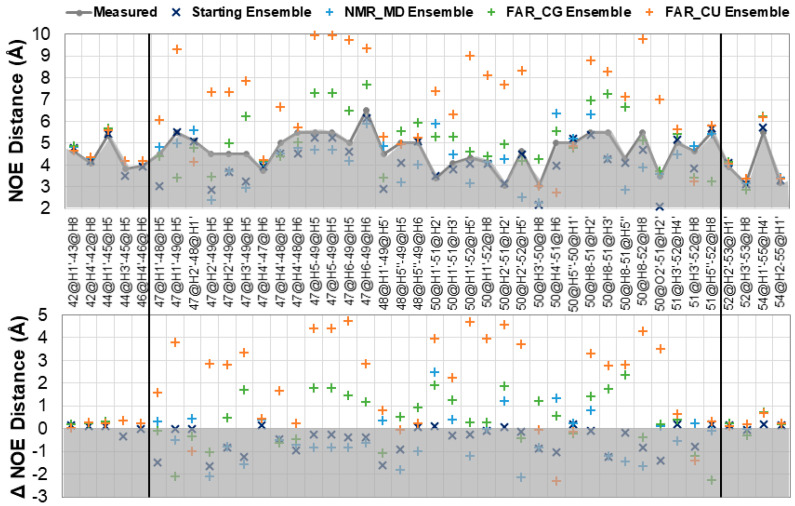
Evaluation of unambiguous nuclear Overhauser effect (NOE) assignments for starting SL2 ensemble, the molecular dynamics ensemble starting from 2L6I models (NMR_MD), the molecular dynamics ensemble starting from FARFAR2 predictions and sorted by central CG hydrogen bond (FAR_CG) or CU hydrogen bond (FAR_CU). Global NMR structure numbering is used for all ensembles in this figure. Top: Average <r^6^> values for ensembles along with the measured NOE data upper limit. Bottom: Deviation of ensemble <r^6^> averages from measured NOE. Grey areas reflect values which fulfil NOEs, as measured values were considered an upper distance limit. NOEs that fall between black bars reflect loop residues.

**Figure 3 ijms-21-06188-f003:**
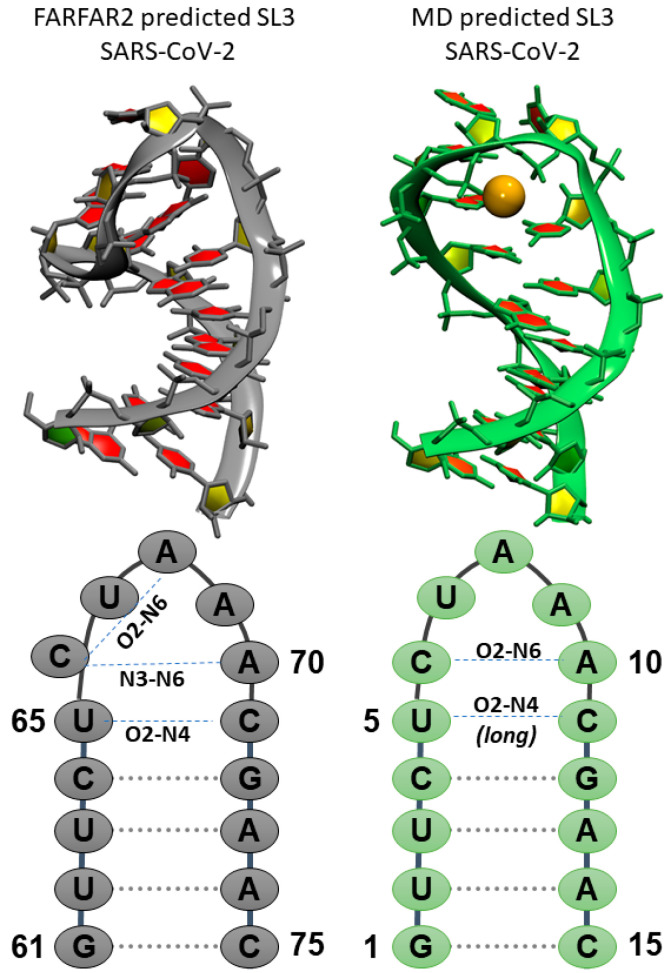
Comparison of predicted SL3 structures from (**left**) de novo tertiary structure prediction FARFAR2 and (**right**) MD top structure prediction for SL3 in SARS-CoV-2. Orange sphere denotes bound K^+^ ion. Sequences, global SARS-CoV-2 numbering and local MD numbering, and deviations in the secondary structure preferences are highlighted in the bottom frames.

**Figure 4 ijms-21-06188-f004:**
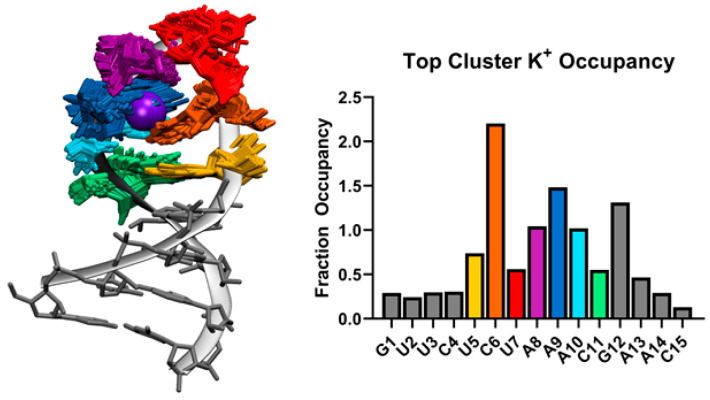
Left: 100 frames from the top cluster from cluster analysis of all SL3 MD simulations overlapped with K^+^ ion density (purple sphere) at 80% occupancy. Right: Fraction occupancy of K^+^ ions for each SL3 residue from the top cluster trajectory. Colors of loop residues UCUAAAC match between the structure and bar graph.

**Table 1 ijms-21-06188-t001:** Chi square goodness of fit for each SL2 ensemble’s NOE violations vs. measured upper values.

	Starting Ensemble	NMR_MD Ensemble	FAR_CG Ensemble	FAR_CU Ensemble
χ^2^	0.11	3.26	8.59	59.84

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
