# Peer review of "Using All-Atom Potentials to Refine RNA Structure Predictions of SARS-CoV-2 Stem Loops"

_ijms, 2020, doi:10.3390/ijms21176188_

Round 1
Reviewer 1 Report
In this manuscript, the author presents a thorough and well-thought out study that could serve as a primer in best-practices for assessing structural models against experimental data using all-atom molecular dynamics simulations.
The work focuses on the detailed 3D structure of two small RNA stem loops of the SARS-CoV2 genome. A recent preprint by the Das group has presented 3D structural models of selected conserved elements of SARS-COV2 using their FARFAR2 fragment-based assembly algorithm constrained by secondary structures resulting from chemical probing experiments. While these are undoubtedly among the most cutting-edge methods currently available, the field of RNA structural prediction is quite young with relatively few major success stories (in contrast to the highly evolved field of protein structural prediction). Given that the stated purpose of the preprint was to provide the scientific community with structural models suitable for virtual drug screening purposes, it is worth asking the question the question posed by this manuscript which is just how accurate can we expect these predictions to be and how would we assess this in absence of experimental structures?
The author focuses mainly on stemloop 2 (SL2) which has the convenient property that it is small enough to reversibly fold via explicit-solvent, all-atom molecular dynamics simulations and there is a previously solved NMR structure of the completely conserved homolog from SARS-COV1. The FARFAR2 prediction for this loop is qualitatively quite different from the NMR structure, with it's predicted CU closing base-pair instead of the NMR observed CG basepair and flipped out U. As the FARFAR2 algorithm only implicitly models stacking without accounting for solvation or ion interactions, the sampling of low-energy loop conformations is commensurably impacted - and it should be pointed out that highly conserved structural loops would the most important potentially drugable targets.
A set of 10 long, unbiased explicit solvent molecular dynamics simulations initialized in the FARFAR2 loop conformations ultimately relax exhibiting fluctuations that break the CU bp and form the NMR-observed CG basepair on the uS timescale, while analogous simulations initialized from the NMR structural ensemble remain essentially unchanged. This tells us that the NMR ensemble is definately closer to the global minimum of the all-atom force-field than the FARFAR2 ensemble although this process seems not to have completely converged at the uS timescale.
The second part of this study stems from the observation that the NMR ensemble is, itself, a structural model resulting from simulated annealing under NOE-derived distance constraints. Therefore, it is desirable to directly calculate agreement of the simulations with the underlying experimental observables, in this case by comparing simulated pairwise distances with the experimentally observed NOEs. In this manner, it is shown the pairwise distance distributions for the FARFAR2 ensemble with CG basepair has much better agreement with experiment than the FARFAR2 ensemble with CU basepair. As a positive control, the MD trajectory initialized from the NMR ensemble has the best agreement with the observed NOEs. With these encouraging results, the author goes on to predict the structure of SL3 and analyze the impact of ions and solvent on it's conformation, which are both details beyond the scope of the FARFAR2 model.
There are several potentially interesting conclusions from this work, (although an explicit conclusion section seems to be missing from the manuscript and should be added). 1) Current all-atom MD RNA force-fields are at least accurate enough to distinguish between a high quality "decoy" structures (from the FARFAR2 knowledge-based model) and a real NMR RNA structure and 2) that long-unbiased MD simulations can potentially be used to further "refine" FARFAR2-results and even correct significant inaccuracies in it's energy model.
I think this study is rigorously done and merits publication with only some minor revisions to address some potential clarifying points below:
- The starting structures from the FARFAR2 clusters are described as the top structure from each of the top 10 clusters - for both the SL2 and SL3 sections, but is was not described how many clusters were there in total and how representative of the whole ensemble are the top 10 clusters? Furthermore, How is “top structure” defined? RMSD to NMR structure? Agreement with NOEs? Lowest FARFAR2 energy?
- In the SL2 simulations, how many of the 10 total FARFAR2 MD simulations ever sampled the central CG hydrogen bond?
- In Figure 2, what was being averaged? Was it the entire ensemble from the whole MD run, or was a part cut out to allow the molecule to reach equilibrium? If it was the entire ensemble, what is the justification for including the times in which the molecule may have been significantly out of equilibrium?
- Why was only 300K temperatures explored? Could slightly elevated temperatures lead to more rapid interconversion of the CU state to the more favorable CG state?
- Were the SL3 MD simulations also run for 1-3μs? Not specified in methods.
- For the SL3 MD clusters, the given top structure is representative of 30% of the ensemble. How does this compare to the second to top structure? Does it contain significantly more structures than any other cluster?
- For both SL2 and SL3, how similar are structures within the same cluster? Perhaps give the total RMSD or the RMSD as compared to the “top structure”?
- Is the top cluster the only group of structures in which the K+ ion is bound? Are there structures that didn’t unravel but were not bound to the K+ ion, and if so, what percentage of structures exhibited this behavior?
Editing Suggestions
- Line 40: what is TRS-B sequence and how it is hypothesized to interact with the TRS-L sequence? (does this implicate existance of a long-range interaction involving the studied highly conserved loops?)
- In Supporting Figures 2 and 3, the labels in the figure legends (C6,G9,U10) don’t match the labels in the captions. Similarly, the text sometimes uses the L1-L4 nomenclature for the loop positions but this is never explained or labelled in the diagrams
- Missing "conclusion" section, see earlier text.
Reviewer 2 Report
In the manuscript 'Using All-Atom Potentials to Refine RNA Structure Predictions of SARS-CoV-2 Stem Loops' the author applied molecular modeling approaches to assess the 3D structure of stem-loop 3 (SL3) in the 5'UTR of SARS-CoV-2. The author compares de-novo prediction data from FARFAR2 and molecular dynamics simulations to experimentally derived NMR structures.
My impression is that the manuscript is well written, and in silico experiments and carefully conducted and described. The overall message of the paper is clear. I have only two minor comments, which I would ask the author to consider and discuss.
- The author mentions in the abstract SHAPE and DMS probing data, however these are not mentioned in the main text. It would be interesting to contextualize these approaches with the present study.
- In the Introduction section, the author gives some background on the biological role of SL3, harboring the TRS-L sequence which is a central component of the discontinuous transcription model of coronaviruses. It would be beneficial to give a broader introduction to the topic, to give readers who are not familiar with CoV discontinuous transcription a better idea why SL3 and TRS regions are important. This could be done as an additional paragraph or in a figure.
